# Executive Function and Processing Speed in Children Living with Sickle Cell Anemia

**DOI:** 10.3390/children10101585

**Published:** 2023-09-22

**Authors:** Stephanie C. Kelleher, Fenella J. Kirkham, Anna M. Hood

**Affiliations:** 1Developmental Neurosciences Unit and Biomedical Research Centre, UCL Great Ormond Street Institute of Child Health, London WC1N 1EH, UK; 2Clinical and Experimental Sciences, University of Southampton, Southampton SO17 1BJ, UK; 3Manchester Centre for Health Psychology, Division of Psychology and Mental Health, University of Manchester, Manchester M13 9PL, UK

**Keywords:** sickle cell disease, executive function, D-KEFS, BRIEF, switching, processing speed, cognitive flexibility, silent infarction, neuropsychology

## Abstract

Executive function and processing speed difficulties are observed in children living with sickle cell anemia (SCA). The influence of processing speed on executive function is not well understood. We recruited 59 children living with SCA and 24 matched controls aged 8–18 years between 2010 and 2016 from clinics in the UK. Children completed tests in processing speed and cognitive flexibility, subdomains of executive function. MRI scans were conducted within one year of testing; oxygen saturation was obtained on the day of testing. Hemoglobin levels were obtained from medical records. Caregivers completed the executive function questionnaire. Hierarchical linear regressions found that hemoglobin, oxygen saturation, age, infarct status, and processing speed were not independent predictors for any model. However, for all cognitive flexibility tests, there was a significant interaction between infarct status and processing speed; children without silent cerebral infarction (SCI) with faster processing speed had better cognitive flexibility. Our findings indicate that, when interpreting executive function difficulties, it is important to account for the relationship between SCI status and processing speed. More research is needed to elucidate the mechanisms, but clinically, including executive function testing as part of clinic visits by embedding psychologists within the healthcare team would appear to be a critical step.

## 1. Introduction

Sickle cell disease (SCD) is a life-threatening inherited red blood cell disorder characterized by the production of abnormal hemoglobin in the red blood cells (RBCs) [1,2]. Sickle cell anemia (SCA) occurs when the homozygous allele is present (HbS) and is often associated with the most clinical severity [3,4]. Approximately 100,000 people are affected in the United States, most with African ancestry, along with 40–50,000 in Europe [5,6] as well as individuals in the Caribbean, South America, the Mediterranean, Saudi Arabia, and India [7].

Hemoglobin S polymerizes in hypoxic conditions, which are also associated with the upregulation of Hypoxia-Inducible-Factor-α genes [8], such as Endothelin-1 [9]. As well as affecting their oxygen-carrying capacity and elasticity, recurrent sickling of RBCs due to polymerized hemoglobin may damage them, leading to hemolysis [10] and the release of free hemoglobin, heme, and extracellular vesicles (microparticles). This is associated with the activation of pathways promoting an inflammatory and procoagulant state, as well as the adhesion of blood cells to the endothelium [11]. The resulting vasculopathy can lead to pain episodes (caused by blockages in the blood vessels due to the rigid RBCs) and significant neurological complications, including silent cerebral infarctions (SCI) and overt stroke [12].

Although there have been significant improvements in the quality of care for patients with SCD in high-income countries such as the US and UK, approximately 25% of children living with SCA will accumulate SCI in the first twenty years of life [13], with SCI occurring in children living with SCA as young as 7 months [14]. Many children living with SCA experience cognitive difficulties compared to typically developing or sibling controls, with the largest differences observed after infarction, although the associations with SCI are less apparent at higher MRI field strengths [15]. Even though stroke and SCI are associated with cognitive difficulties [16], children with no detectable infarction or abnormalities on an MRI also experience challenges which have been linked to insufficient delivery of oxygen to the brain (e.g., anemia-induced tissue hypoxia) [17,18].

Lower oxygen saturation and hemoglobin levels are related to white matter (WM) damage in people living with SCA [19]. As the brain tightly regulates oxygen delivery, and approximately 98% of oxygen is transported by hemoglobin, lower hemoglobin levels in people living with SCA ultimately reduce the oxygen-carrying capacity of the blood. To compensate for this reduction, the brain elevates the baseline cerebral blood flow [20,21] via cerebral autoregulation (dilation of the cerebral arterioles) to maintain expected global oxygen delivery. Whilst oxygen delivery to grey matter is relatively preserved, deep WM structures may be hypoxic [22] and hypoperfused proportionally to the severity of anemia [23], leading to metabolic and hemodynamic stress.

Cognition encompasses multiple domains, including visual–spatial abilities, language, and memory [24]. However, previous research on SCD has most often only focused on the intelligence quotient (IQ) as a measure of cognition [16,25], even though it has become increasingly clear that children living with SCA consistently have lower scores in the domains of processing speed [26,27] and executive function [28,29,30]. In fact, adults living with SCA have been shown to perform similarly to controls on tests of intelligence once processing speed has been taken into account statistically [31]. Processing speed measures efficient management, absorption, and rapid response to the presented information [32]. Executive function includes skills that represent the capacity to plan and meet goals, maintain focus despite distractions, engage in goal-directed behavior, display self-control, and follow multiple-step directions [33,34]. SCA infarction often occurs in the frontal lobe regions [35], which are greatly affected by hypoxia (reduced tissue oxygenation) and hypoxemia (decrease in the partial pressure of oxygen in the blood) [13], and these regions are vital in facilitating executive function [35,36,37,38].

Cognitive flexibility is a critical aspect of executive function and reflects the ability to adapt behaviors in response to environmental changes [34,39,40]. Previous research has demonstrated difficulties in cognitive flexibility in infants and children living with SCA [41], specifically related to challenges in task switching [42] and sorting [35]. Children living with SCA performed poorly on organization tasks (e.g., assembling objects) and information-processing tasks [43]. Moreover, switching scores were poorer in children living with SCA [41], and 45% of children living with SCA had decreased sorting test performance compared to controls [44].

Given that children living with SCA experience processing speed and executive function (cognitive flexibility) difficulties [45], understanding of how these cognitive domains interact along with the roles of infarction and reduced oxygen delivery in SCA (i.e., oxygen saturation, hemoglobin) is urgently needed. Despite advances in care and increased availability of disease-modifying treatments [46,47,48], previous research has demonstrated that executive function problems in children living with SCD are related to sleep difficulties [49], reduced social skills [41], challenges with pain coping and quality of life [50,51,52], and persistent pain [53]. Critically, adolescents with SCD exhibit significant challenges with managing their medical care and decision-making [54,55], likely because these decisions require executive function (remembering information) and language comprehension (understanding and reasoning) skills. Cognitive difficulties typically increase with age in the general population [56], and executive function deficits translate into functional limitations (e.g., reduced employment) in daily life for adult patients living with SCD [57].

Given the academic, medical, and functional needs to improve executive function for children living with SCD, and in view of childhood being a critical window, this study investigated whether reduced oxygen delivery (e.g., poorer oxygen saturation, lower hemoglobin) and infarct status predicted cognitive flexibility difficulties. Additionally, we endeavored to determine the underlying role of processing speed and whether there is an interaction between processing speed and infarct status. Finally, we evaluated whether a measure of caregiver-reported executive function is related to these executive function domains. Previous studies have included participants across sickle cell genotypes (e.g., HbSC) without matched controls; therefore, this study aimed to address these limitations and to aid medical providers and psychologists working with children living with SCA by providing further insight into underlying cognitive and brain mechanisms, as well as to help inform interventions and mitigate executive function challenges for children living with SCA.

## 2. Materials and Methods

Ethical approval was granted by the Southampton and Southwest Hampshire Research Ethics Committee (reference 09/NR/17) according to NHS guidelines. Written informed consent was given by (1) parents/guardians for children under 16 years and (2) participants over 16 years of age. Caregivers and children were allowed at least twenty-four hours to review the relevant information sheets and decide whether they would participate.

### 2.1. Participants

Children and adolescents with SCA were recruited from Whittington, Southampton General, and North Middlesex hospitals. Inclusion criteria for children living with SCA were (1) HbSS genotype, (2) aged between 8 and 18 years, and (3) attending an English-speaking school in the United Kingdom. Exclusion criteria included a history of brain trauma or epilepsy. In addition, we also recruited familial and sibling ethnicity-matched controls within the same age range using the same methods.

### 2.2. Procedures

Consultants discussed the study recruitment with the caregivers of the children living with SCA during sickle clinics, or contacted them via phone or letter. Families who consented to the study completed cognitive testing at the clinic visit or came to the university campus. All testing was completed in a quiet room and took approximately 2 hours; tests were conducted between 2010 and 2016. Additionally, pulse oximeter (Masimo Pronto-7) measurements (i.e., oxygen saturation) were obtained on the day of testing and took 5 min to complete. Steady-state hemoglobin levels taken at the closest clinical visit were obtained from the patient’s medical records. Caregivers of children under 16 completed the executive function questionnaire while their child was being tested.

### 2.3. Materials

#### 2.3.1. Delis–Kaplan Executive Function System

Cognitive flexibility was assessed using three of the nine subtests from the Delis–Kaplan Executive Function System (D-KEFS) assessment battery [58]: The Trail-Making Test, Number–Letter Switching Condition (i.e., processing multiple concepts simultaneously); the Color–Word Inference Test, Switching Condition, specifically, verbal inhibition (scores are based on the time taken); and the total achieved score on the Tower Test measured cognitive flexibility. The Trail-Making Test, Visual Scanning condition measured processing speed. Lower scores signify more executive function difficulties (Mean = 10; Standard Deviation = 2).

#### 2.3.2. The Behavior Rating Inventory of Executive Function

The Behavior Rating Inventory of Executive Function (BRIEF) is a standardized questionnaire comprising 86 items that reflect the behaviors of children related to observed executive function in the home and school environments [59]. The caregiver-reported questionnaire was used in the present study. The BRIEF comprises eight subscales: Inhibit, Shift, Emotional Control, Initiate, Working Memory, Plan/Organize, Organization of Materials, and Monitor. The first three subscales form the Behavioral Regulation Index (BRI), and the last five subscales make up the Metacognition Index (MI); the BRI and MI are then summed to create a Global Executive Composite (GEC) score. The GEC T-score was used in analyses, with higher scores signifying poorer executive function (Mean = 50; Standard Deviation = 10).

#### 2.3.3. MRI Acquisition

MRI data were acquired in all subjects using a 1.5 T Siemens Magnetom Avanto (Siemens, Erlangen, Germany) with 40 mT/m gradients and a 32-channel receive head coil. For lesion diagnosis, we acquired a T2-weighted turbo spin echo sequence (repetition time = 4920 ms; echo time = 101 ms, voxel size = 0.7 × 0.6 × 4.0 mm). MRI scans were conducted within one year of cognitive testing, and two experienced neuroradiologists read all participants’ T2-weighted MRI blinded to disease status.

### 2.4. Statistical Analyses

The statistical package SPSS version 25 was utilized for data cleaning and management [60]. The statistical package R was utilized to analyze all of the data [61]. Summary analyses described children living with SCA and the familial and sibling control sample. Chi-squared, Welch’s independent samples *t*-tests, or ANOVAs assessed for group differences. Pearson correlations assessed relationships between executive function tests and measures, and hierarchical linear regressions, in which the order of the predictor variables is based on theory and decided a priori by the researcher, assessed whether medical variables and processing speed predicted executive function. Correlations and linear regression models were only conducted for the children living with SCA. Initially, oxygen saturation and hemoglobin were included as predictors in all of the regression models. However, as they were not significant predictors for any test of cognitive flexibility (all ps > 0.3), they were not included in subsequent analyses, to increase statistical power. Five children had previously experienced a mild stroke (determined by their neurologist), but they had made a complete neurological recovery, and their executive function data were similar (i.e., <0.5 SD) to those of children living with SCI; therefore, they were grouped with children living with SCI for all analyses.

## 3. Results

Fifty-nine Black children of African or Caribbean heritage living with SCA were included in the present study. We also included 24 familial or sibling ethnicity-matched controls. Children living with SCA were significantly older than the controls (1-year mean difference); however, there were similar numbers of females and males across groups (Table 1).

There were no differences in hemoglobin or oxygen saturation levels between children living with SCA with or without SCI (Table 2). Children living with SCA generally had lower scores on tests of cognitive flexibility than the controls, with scores 1 to 4.5 below the normative mean (M = 10). Children living with SCA and SCI demonstrated statistically lower scores on the D-KEFS Tower Test and Color–Word Switching Condition than controls. There were no differences between those living with SCA on the Trail-Making Switching or Visual Scanning Conditions. BRIEF GEC mean T-scores for all groups were within normal limits. However, caregivers reported more executive function difficulties for children living with SCI (Table 2).

A similar percentage of controls and children living with SCA were within the clinical range on the BRIEF (*p* > 0.05) (Figure 1).

We next assessed the relationship between tests of cognitive flexibility and the GEC T-scores on the BRIEF. We found that two switching tests (Trail-Making and Color–Word) were significantly negatively correlated with the GEC T-scores on the BRIEF. The Tower Test was also correlated with the BRIEF, but the result did not reach statistical significance. Tests of cognitive flexibility other than the Tower and Color–Word Switching Condition were significantly related (see Table 3).

We conducted four individual regression analyses with independent predictors of age, infarct status, and processing speed (Trail-Making Visual Scanning Condition). In our first model, with the Trail-Making Switching Condition as our dependent variable, we found that the overall model was statistically significant, *F* (4, 48), 5.31, *r*^2^ = 0.31 *p* = 0.001. We found that no independent predictors significantly predicted Trail-Making Switching. However, there was a significant interaction between infarct status and Trail-Making Visual Scanning (*p* = 0.04), such that children living with SCA without an infarct and with higher processing speed scores performed better on this cognitive flexibility switching test.

In our second model, with the Color–Word Inference Switching Condition test as our dependent variable, we found that the overall model was not statistically significant, *F* (4, 44), 1.76, *r*^2^ =.14 *p* = 0.15. We found that no independent predictors significantly predicted the Color–Word Inference Switching Condition. However, there was a significant interaction between infarct status and Trail-Making Visual Scanning (*p* = 0.05), such that children living with SCA without an infarct and with higher processing speed scores performed better on this cognitive flexibility switching test.

In our third model, with the Tower Test as our dependent variable, we found the overall model was statistically significant, *F* (4, 45), 4.70, *r*^2^ = 0.30 *p* = 0.003. We found that no independent predictors significantly predicted the Tower Test. However, there was a significant interaction between infarct status and Trail-Making Visual Scanning (*p* = 0.04), such that children living with SCA without an infarct and with higher processing speed scores performed better on this cognitive flexibility planning test.

In our final model, with the BRIEF GEC T-score as our dependent variable, we found that the overall model was not statistically significant *F* (4, 31), 0.98, *r*^2^ = 0.002. *p* = 0.43, and there were no independent predictors or interaction between infarct status and processing speed (see Table 4).

## 4. Discussion

Infarction and chronically poor oxygen delivery to the brain for children living with SCA can result in cognitive difficulties with long-term medical, academic, and functional consequences. Therefore, it is critical to determine the etiology and mechanisms of cognitive challenges to inform future interventions. For many years, research in children living with SCA focused on IQ [62]; however, more recent research has found that the most profound cognitive difficulties are the potentially modifiable domains of executive function and processing speed. It is also likely that the lower IQ values observed were driven by executive function and processing speed challenges. The present study assessed whether lower executive function and processing speed scores were observed for children living with SCA with and without SCI, and if processing speed and infarct status predicted executive function.

Our findings support that children who are living with SCA with SCI generally had lower executive function and processing speed scores than sibling and ethnicity-matched control samples. This finding is generally supported in the literature [16]. The robust correlations between our caregiver-reported measure of executive function and the cognitive flexibility tests indicate that the BRIEF is a suitable clinical screening measure for busy sickle cell clinics that can identify those children living with SCA who may need additional neuropsychological testing to provide a more holistic picture with regards to executive function difficulties. We also found that age, hemoglobin, and oxygen saturation were not significant predictors when considering executive function performance on switching and planning tests in regression models only including children living with SCA. As scores on executive function are standardized for age, this may explain why age was not a predictor. Additionally, our sample had minimal hemoglobin and oxygen saturation variability, and we did not have many children with very low levels. Nevertheless, future work could look at longitudinal hemoglobin and oxygen saturation levels to clearly indicate levels over time rather than only in a single measurement.

Of particular interest, we found that, for all tests of cognitive flexibility, age, infarct status, and processing speed were not independent predictors. However, there was a significant interaction between infarct status and processing speed for all models, such that children living with SCA who *had not* experienced an infarction and had faster processing speed performed better on cognitive flexibility tests. This advantage was not present for children living with SCA who *had* experienced an infarction. More research is needed to elucidate the mechanisms that underlie these differences, but assessing whether infarction in the visual pathway specifically affects visual processing would appear to be a plausible next step. One important note is that our measure of processing speed is the first condition (Visual Scanning) in the Trail-Making Test, so it might seem unsurprising that it was related to the fourth condition (Switching). However, the same measure of processing speed also predicted two different cognitive flexibility tests of switching (Color–Word Interference) and planning (Tower Test), suggesting that the relationship is not test-dependent, but instead is executive function domain-dependent. Specifically, processing speed seems particularly related to cognitive flexibility in children living with SCA.

There has been a proliferation of treatment options in the past few years for children living with SCA [63], which have the potential to improve oxygen delivery and, thus, executive function and processing speed. There is preliminary evidence that executive function is amenable to change for children living with SCA through blood transfusion [29] and cognitive training [30,64]. To gain greater clarity, executive function and processing speed tests should be included as endpoints in clinical trials, as they represent domains that, if improved, would result in clinically meaningful change in the daily lives of children living with SCA [65]. Including executive function and processing speed as endpoints in clinical trials would lead to a better understanding, with findings then applied to improve academic attainment, social functioning, and quality of life for children living with SCA [41,52].

Parents, clinicians, and educational professionals often underestimate the cognitive delay within the SCA population [66]. Promoting executive function testing early in child development could ameliorate achievement gaps [41]. Our findings have shown that utilizing the caregiver-reported BRIEF and subtests from the D-KEFS battery would be ideal as screening measures to quickly identify executive function deficits in patients with SCA. Clinicians only have a short time with the children; thus, effectively using the time available is imperative. One way to best utilize this time is to have the child complete processing speed and cognitive flexibility tests (approximately 10–15 min to complete) while the parent/caregiver completes the BRIEF measure. The BRIEF-II has a short form that could be completed in a shorter timeframe and has demonstrated applicability in the SCD population [67]. Additionally, the NIH Toolbox is a brief standardized screening battery [68] that has been shown to be successful as a cognitive screening measure that can be used clinically in this pediatric patient population [28,29]; however, there is still a lack of evidence for its clinical use.

Screening tests could determine those patients who need full neuropsychological assessments with ecological performance-based tests akin to everyday tasks and require cognitive flexibility, allowing neuropsychologists to develop practical and individualized interventions to support children at home and school. This practice has proved successful in a large SCD clinic in the US, where psychologists are embedded within the multidisciplinary team [69], and could be adopted in lower-income countries, albeit with tests normalized specifically for local populations. We do not currently recommend tele-neuropsychology (i.e., remote cognitive testing) for children living with SCA, as, although the current small literature base demonstrates reliability with in-person testing for some cognitive domains, these data are based on evidence that has limited reporting of sex-assigned birth, racialized identity, and ethnicity, and few studies examined processing speed [70].

### Limitations

The strengths of our study lie in the size and homogeneity of this cohort, which overcome the limitations of previous studies that have included children living with SCD with differing genotypes (e.g., HbSC). Control participants were demographically matched; therefore, the likelihood of previously reported confounding variables (e.g., home environment) is reduced. Despite these strengths, data were collected from the South of England only, so they may not be generalizable to other geographic regions, and we could not assess the influence of disease-modifying treatments on executive function. Further, socioeconomic status (SES) was not recorded for either group, which makes it challenging to know the impact of deprivation on test scores. Children with SCA often face SES disadvantages, which have been shown to negatively impact executive function [71,72]. Another limitation of our study is that, although imaging data were available, we could only assess the presence or absence of infarction. The presence or absence of SCI may not be the most sensitive biomarker for detecting functionally significant pathology [15].

## 5. Conclusions

Our study found that, in line with previous work, children living with SCA have executive function and processing speed difficulties when assessed using tests of cognitive flexibility and caregiver-reported questionnaires. Previous research has shown that processing speed underlies discrepancies in IQ [31]. Our results demonstrate that this relationship is present for executive function, as well. However, the benefit of improved executive function through faster processing speed was only available for those children who had not experienced an SCI. The mechanisms through which this differential advantage is conferred require further inquiry. However, neuropsychological practice and both observational and clinical trial research should focus on the potentially modifiable [73] domains of executive function and processing speed. Identifying challenges early in development through screening [74] could mitigate later challenges and allow for support at school and home.

## Figures and Tables

**Figure 1 children-10-01585-f001:**
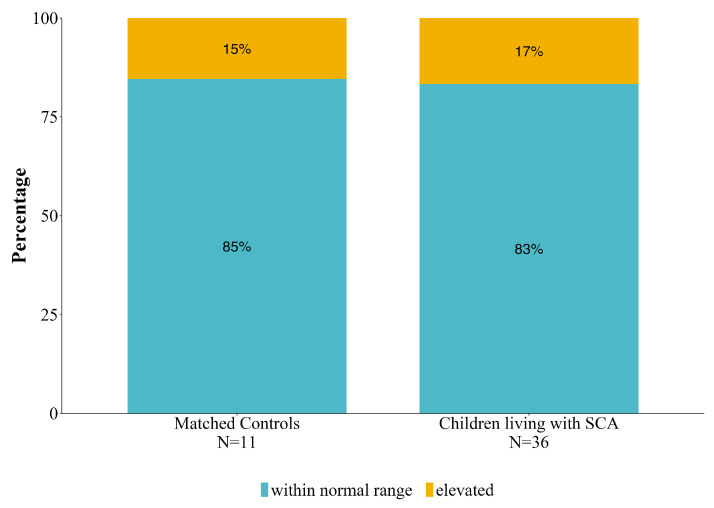
Percentages of control and children living with SCA within normal range and with elevated BRIEF executive function scores.

**Table 1 children-10-01585-t001:** Participant characteristics.

Characteristics	Sickle Cell Anemia *n* = 59	Controls *n* = 24	*p*-Value
Age (years)			
Mean (SD)	12.97 (2.70)	11.68 (2.16)	0.04
Range	8–18 years	8–16 years	
Sex assigned at birth			
Female	34 (57.6%)	10 (41.7%)	0.28
Male	25 (42.4%)	14 (58.3%)	
Ethnicity			
African or Caribbean Heritage (Black)	59 (100%)	24 (100%)	–
Genotype			
HbSS	59 (100%)	0 (0%)	–
HbAA and HbAS	–	24 (100%)	–
Infarct Status			
SCI or mild stroke	17 (28.8%)	–	–
No infarct	39 (66.1%)	–	–
Unknown	3 (5.1%)	–	–

Note: SD = Standard Deviation; SCI = Silent Infarct. Chi-squared tests assessed for differences between groups.

**Table 2 children-10-01585-t002:** Descriptive statistics for all study variables assessed by infarct status for children living with SCA and including matched controls.

Variables	SCI (*n* = 17)	No SCI (*n* = 39)	Controls (*n* = 24)	*p*-Value
Hemodynamic markers				
Hemoglobin levels	8.31 (1.36)	8.47 (1.19)	–	0.70
Oxygen saturation	96.79 (1.85)	96.45 (3.09)	–	0.71
D-KEFS Cognitive Flexibility				
Tower Test (Achievement Score)	8.00 (1.54)	9.18 (2.35)	10.05 (1.90)	0.01
Trail-Making Switching Condition	5.53 (3.62)	7.45 (3.85)	6.71 (3.45)	0.24
Color–Word Switching Condition	6.94 (2.99)	8.12 (3.57)	9.94 (2.79)	0.04
BRIEF				
GEC (T-score)	58.40 (10.07)	52.81(9.91)	49.46 (9.90)	0.14
Processing Speed				
Trail-Making Scanning Condition	8.56 (3.24)	8.69 (3.85)	9.19 (3.34)	0.84

Note: SD = Standard Deviation; SCI = Silent Infarct; BRIEF = Behavior Rating Inventory of Executive Function; GEC = General Executive Composite. Welch’s independent samples t-tests assessed for differences between groups.

**Table 3 children-10-01585-t003:** Correlations between caregiver-reported executive function (BRIEF) and three tests of cognitive flexibility.

Variable	1	2	3
1. BRIEF GEC T-Score			
2. Tower Test (Achievement Score)	−0.33		
	[−0.60, 0.01]		
3. Trail-Making Switching Condition	−0.41 *	0.36 *	
	[−0.66, −0.08]	[0.09, 0.58]	
4. Color–Word Switching Condition	−0.45 **	0.12	0.51 **
	[−0.68, −0.13]	[0.02, 0.51]	[0.35, 0.72]

Note. Values in square brackets indicate the 95% confidence interval for each correlation. * indicates *p* < 0.05. ** indicates *p* < 0.01. BRIEF = Behavior Rating Inventory of Executive Function; GEC = Global Executive Composite.

**Table 4 children-10-01585-t004:** Hierarchical linear regressions for the D-KEFS Cognitive flexibility tests (Trail-Making Switching, Color–Word Interference, and Tower Planning) and caregiver-reported executive function (BRIEF GEC T-score) as the dependent variables.

Predictor	*b*	*b* 95% CI [LL, UL]	*sr* ^2^	*sr*^2^ 95% CI [LL, UL]	*p*	Partial Eta-Squared	Fit
**Trail-Making Switching**							
(Intercept)	10.72 *	[1.11, 20.32]			0.03		
Age	−0.23	[−0.61, 0.15]	0.02	[−0.04, 0.09]	0.22	0.03	
Infarct Status	−5.19	[−12.51, 2.12]	0.03	[−0.05, 0.11]	0.16	0.06	
Trail-Making Visual Scanning	−0.21	[−0.91, 0.50]	0.00	[−0.03, 0.04]	0.56	0.20	
Infarct Status * Visual scanning (Processing Speed)	0.77 *	[0.01, 1.53]	0.06	[−0.05, 0.17]	0.04	0.08	
							*R*^2^ = 0.307 **
							95% CI [0.06, 0.44]
**Color–Word Interference Switching**							
(Intercept)	14.42 **	[6.21, 22.63]			0.001		
Age	−0.27	[−0.67, 0.12]	0.04	[−0.06, 0.14]	0.17	0.04	
Infarct Status	−4.49	[−10.66, 1.68]	0.04	[−0.06, 0.15]	0.15	0.02	
Trail-Making Visual Scanning	−0.45	[−1.02, 0.12]	0.05	[−0.06, 0.16]	0.12	0.001	
Infarct Status * Visual scanning (Processing Speed)	0.64	[−0.02, 1.30]	0.07	[−0.06, 0.21]	0.05	0.08	
							*R*^2^ = 0.139
							95% CI [0.00, 0.27]
**Tower Planning**							
(Intercept)	11.27 **	[6.85, 15.69]			<0.001		
Age	−0.18	[−0.39, 0.03]	0.05	[−0.05, 0.15]	0.09	0.06	
Infarct Status	−2.46	[−5.78, 0.86]	0.03	[−0.05, 0.12]	0.14	0.03	
Trail-Making Visual Scanning	−0.06	[−0.37, 0.24]	0.00	[−0.02, 0.03]	0.68	0.15	
Infarct Status * Visual scanning (Processing Speed)	0.37 *	[0.01, 0.72]	0.07	[−0.05, 0.19]	0.04	0.09	
							*R*^2^ = 0.295 **
							95% CI [0.05, 0.43]
**BRIEF GEC T-score**							
(Intercept)	55.58 **	[26.05, 85.11]			0.001		
Age	−0.09	[−1.70, 1.52]	0.00	[−0.01, 0.01]	0.91	0.0004	
Infarct Status	3.89	[−17.65, 25.42]	0.00	[−0.03, 0.04]	0.71	0.06	
Trail-Making Visual Scanning	0.47	[−1.63, 2.56]	0.01	[−0.04, 0.05]	0.65	0.02	
Infarct Status * Visual scanning (Processing Speed)	−1.09	[−3.43, 1.24]	0.03	[−0.07, 0.12]	0.35	0.03	
							*R*^2^ = 0.112
							95% CI [0.00, 0.24]

Note. A significant *b*-weight indicates that the semi-partial correlation is also significant. *b* represents unstandardized regression weights. *sr*^2^ represents the semi-partial correlation, squared. *LL* and *UL* indicate the lower and upper limits of a confidence interval, respectively. * indicates *p* < 0.05. ** indicates *p* < 0.01.

## Data Availability

The data that support the findings of this study are available from the corresponding author, F.J.K., upon reasonable request.

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
