# Peer review of "Executive Function and Processing Speed in Children Living with Sickle Cell Anemia"

_children, 2023, doi:10.3390/children10101585_

Round 1

Reviewer 1 Report

In this paper, the aim was to investigate whether reduced oxygen delivery and infarct status predicted cognitive flexibility difficulties, and to determine the role of processing speed and its interaction with infarct status. The main contribution to the literature is the finding that for executive functioning, an interaction exists between infarct status and processing speed. Manuscript is clearly written and relevant to the field of sickle cell disease. It addresses a major gap in knowledge, as cognitive deficiencies are encountered at an increasing rate from childhood to adulthood. Incorporating executive functioning tests early and a standard of care, can help identify patients with difficulties and mitigate long-term comorbidities associated with their deficiencies.

Comments below:

1)     Pg 1: In the abstract, the abbreviation SCI is used without first spelling out.

2)     Pg 2: Line 50. The abbreviation WM is used without first spelling out.

Author Response

In this paper, the aim was to investigate whether reduced oxygen delivery and infarct status predicted cognitive flexibility difficulties and to determine the role of processing speed and its interaction with infarct status. The main contribution to the literature is the finding that for executive functioning, an interaction exists between infarct status and processing speed. The manuscript is clearly written and relevant to the field of sickle cell disease. It addresses a major gap in knowledge, as cognitive deficiencies are encountered at an increasing rate from childhood to adulthood. Incorporating executive functioning tests early, and a standard of care can help identify patients with difficulties and mitigate long-term comorbidities associated with their deficiencies.

Response: We thank the reviewer for these comments

  1. Pg 1: In the abstract, the abbreviation SCI is used without first spelling out.

This has been corrected.

  1. Pg 2: Line 50. The abbreviation WM is used without first spelling out.

This has been corrected.

Reviewer 2 Report

Stephanie Kelleher  and colleagues report on functional cognitive study in children with Sickle Cell Disease. This paper is of interest and very actual. Overall is well written and I believe will achieve good citations.

My reccomandation is: accept it if appropriately modified.

MAJOR

- Please note this bit

line 243 244

"We also found that age, hemoglobin, and oxygen saturation 243 were not significant predictors when considering executive function performance on 244 switching and planning tests" 

Authors need to state clearly that these results may be bias and not true because the control group was taken from sibling. I believe this is the most critical point of all this article. This should be clearly stated even in the abstract.

MINOR CORRECTIONS REQUIRED:

ABSTRACT

- when dividing the word "record" check syllabic division

- what authors mean under "Hierarchical"  linear regressions? please explain

INTRODUCTION:

- define acronym "WM"  

- add this Ref after Ref 2:

Insight into the complex pathophysiology of sickle cell anaemia and possible treatment.

Piccin A, Murphy C, Eakins E, Rondinelli MB, Daves M, Vecchiato C, Wolf D, Mc Mahon C, Smith OP.Eur J Haematol. 2019 Apr;102(4):319-33

- add this Ref after Ref 5: 

Sickle cell disease and dental treatment.

Piccin A, Fleming P, Eakins E, McGovern E, Smith OP, McMahon C.J Ir Dent Assoc. 2008 Apr-May;54(2):75-9   - after "to metabolic and hemodynamic stress " need to create 2-3 paragraphs summarising the key mechanisms on vaso-occlusione and endothelial damage mechanisms in SCD, explaining the role of adhesion molecules, Nitric Oxide and Protein C consumption, Microparticles generation etc use these references: Circulating microparticles, protein C, free protein S and endothelial vascular markers in children with sickle cell anaemia. Piccin A, Murphy C, Eakins E, Kunde J, Corvetta D, Di Pierro A, Negri G, Guido M, Sainati L, Mc Mahon C, Smith OP, Murphy W.J Extracell Vesicles. 2015 Nov 23;4:28414 The 'scintilla' starting vaso-occlusion in sickle cell disease. Piccin A, Magzoub I, Hervig T.Br J Haematol. 2023 May;201(3):379-380 Protein C and free protein S in children with sickle cell anemia. Piccin A, Murphy C, Eakins E, Kinsella A, McMahon C, Smith OP, Murphy WG.Ann Hematol. 2012 Oct;91(10):1669-71.   

- page 2 line 70 "hypoxia and hypoxemia ", please use only one of these words or is confusing 

- define what you mean under "cognitive flexibility " or delete it 

- delete Figure 1 . Is redundant, instead  provide a p value showing ns difference between 2 groups

- page 7 line 238, correct/replace "extant literature "

- Please note this bit "We also found that age, hemoglobin, and oxygen saturation 243 were not significant predictors when considering executive function performance on 244 switching and planning tests" 

Authors need to state clearly that this result may be bias and not true because the control group was taken from sibling. I believe this is the most critical point of this article

Author Response

Stephanie Kelleher and colleagues report on functional cognitive study in children with Sickle Cell Disease. This paper is of interest and very actual. Overall is well written, and I believe will achieve good citations.

My recommendation is: accept it if appropriately modified.

Response: We thank the reviewer for these comments

MAJOR

  1. Please note this bit on line 243 244

"We also found that age, hemoglobin, and oxygen saturation were not significant predictors when considering executive function performance on switching and planning tests" The authors need to state clearly that these results may be biased and not true because the control group was taken from siblings. I believe this is the most critical point of all this article. This should be clearly stated even in the abstract.

Response: We believe that the authors are referring to our findings that age, hemoglobin, and oxygen saturation were not significant predictors in our regression models. As we state on line 156, page 4, regression models do not include siblings, so this would not introduce any bias whether or not siblings are controls. To make this clearer, the sentence in the Discussion now reads as follows:

“We also found that age, hemoglobin, and oxygen saturation were not significant predictors when considering executive function performance on switching and planning tests in regression models only including children living with SCA.

MINOR CORRECTIONS REQUIRED:

  1. ABSTRACT

When dividing the word "record," check syllabic division

Response: This division was not created by the authors, it occurred via the Children's template

  1. What do authors mean under "Hierarchical"  linear regressions? please explain

Response: Hierarchical linear regressions are a common statistical analysis for regressions in which the order of the predictor variables is based on theory and decided a priori by the researcher. We have added the following to the Statistical Analysis section.

“…hierarchical linear regressions, in which the order of the predictor variables is based on theory and decided a priori by the researcher…”

INTRODUCTION:

  1. Define the acronym “WM”

Response: This has been added

  1. Add reviewer citations

Response: Given the editors' concerns about self-citation, we are concerned about adding references from the reviewer’s papers that do not directly relate to the current work. However, the following relevant references from the reviewer and other authors have been added to the manuscript:

Piccin A, Murphy C, Eakins E, Kunde J, Corvetta D, Di Pierro A, Negri G, Guido M, Sainati L, Mc Mahon C, Smith OP, Murphy W. Circulating microparticles, protein C, free protein S and endothelial vascular markers in children with sickle cell anaemia. J Extracell Vesicles. 2015 Nov 23;4:28414. doi: 10.3402/jev.v4.28414. PMID: 26609806; PMCID: PMC4658688.

Piccin, A., Murphy, C., Eakins, E., Rondinelli, M. B., Daves, M., Vecchiato, C., ... & Smith, O. P. (2019). Insight into the complex pathophysiology of sickle cell anaemia and possible treatment. European journal of haematology, 102(4), 319-330.

An R, Man Y, Cheng K, Zhang T, Chen C, Wang F, Abdulla F, Kucukal E, Wulftange WJ, Goreke U, Bode A, Nayak LV, Vercellotti GM, Belcher JD, Little JA, Gurkan UA. Sickle red blood cell-derived extracellular vesicles activate endothelial cells and enhance sickle red cell adhesion mediated by von Willebrand factor. Br J Haematol. 2023 May;201(3):552-563. doi: 10.1111/bjh.18616. Epub 2023 Jan 5. PMID: 36604837; PMCID: PMC10121869.

Pedrosa AM, Lemes RPG. Gene expression of HIF-1α and VEGF in response to hypoxia in sickle cell anaemia: Influence of hydroxycarbamide. Br J Haematol. 2020 Jul;190(1):e39-e42. doi: 10.1111/bjh.16693. Epub 2020 Apr 30. PMID: 32352161.

  1. Need to create 2-3 paragraphs summarising the key mechanisms of vaso-occlusion and endothelial damage mechanisms in SCD, explaining the role of adhesion molecules, Nitric Oxide and Protein C consumption, Microparticles generation etc use these references: Circulating microparticles, protein C, free protein S and endothelial vascular markers in children with sickle cell anaemia.

We have included a paragraph with the above references, which reads as follows:

“Hemoglobin S polymerizes in hypoxic conditions, which are also associated with the upregulation of Hypoxia-Indicible-Factor-α genes [6], such as Endothelin-1 [7]. As well as affecting the oxygen-carrying capacity and elasticity, recurrent sickling of RBCs due to polymerized hemoglobin may damage them, leading to hemolysis [8] and the release of free hemoglobin, heme and extracellular vesicles (microparticles). This is associated with the activation of pathways promoting an inflammatory and procoagulant state and the adhesion of blood cells to the endothelium [9]. The resulting vasculopathy can lead to pain episodes (caused by blockages in the blood vessels due to the rigid RBCs), and significant neurological complications, including silent cerebral infarctions (SCI) and overt stroke [10].”

  1. Page 2, line 70, "hypoxia and hypoxemia ", please use only one of these words or is confusing. 

Response: Hypoxia and hypoxemia are not synonyms. Hypoxemia is defined as a decrease in the partial pressure of oxygen in the blood; whereas, hypoxia is defined as a reduced level of tissue oxygenation. Therefore, both terms remain, but were have added the following:

“SCA infarction often occurs in the frontal lobe regions, which are greatly affected by hypoxia (reduced tissue oxygenation) and hypoxemia (decrease in the partial pressure of oxygen in the blood) [7], and these regions are vital in facilitating executive function [27,28].”

  1. Define what you mean under "cognitive flexibility " or delete it. 

Response: Cognitive flexibility is defined on line 14 and page 1 in the abstract as a subdomain of executive function. Cognitive flexibility is also defined in more detail on line 73, page 2.

  1. Delete Figure 1. Is redundant, instead provide a p-value showing the ns difference between 2 groups

Response: We are unclear on how the figure is redundant as the data is not previously presented. Figures are the quickest way to communicate information that would be complicated to explain in text. We have, however, added the p-value as requested.

  1. Page 7, line 238, correct/replace "extant literature "

Response: The word extant has been removed.

  1. Please note this bit "We also found that age, hemoglobin, and oxygen saturation 243 were not significant predictors when considering executive function performance on 244 switching and planning tests" Authors need to state clearly that this result may be biased and not true because the control group was taken from a sibling. I believe this is the most critical point of this article.

Response: This comment is a repeat of the first comment from the reviewers. Please see our response to that question indicating that controls were not included in regression models.

Reviewer 3 Report

The article “Executive function and processing speed in children living with sickle cell anemia” is very interesting and worth publishing but I have following comments/suggestions,

1. (2.3.2) More information is required for better understanding of The Global Executive Composite (GEC) t-score.

2. (2.4) more information should be provided in this section so that readers may get an idea where the authors have utilized SPSS and where R.

3. Table 1 & 2 Which statistical test was used to calculate the reported p-values.

4. Figure 1. The percentages within the bars are very small and the authors must increase their font size.  

Author Response

The article “Executive function and processing speed in children living with sickle cell anemia” is very interesting and worth publishing, but I have the following comments/suggestions.

Response: We thank the reviewer for these comments.

  1. More information is required for a better understanding of The Global Executive Composite (GEC) t-score.

Response: We appreciate this comment. More information about the BRIEF and GEC has been added. The manuscript now reads as follows:

“The Behavior Rating Inventory of Executive Function (BRIEF) is a standardized questionnaire consisting of 86 items that reflect behaviors of children related to observed executive function in the home and school (Gioia, Isquith, Guy, & Kenworthy, 2000). The caregiver-reported questionnaire was used in the present study. The BRIEF comprises eight subscales: Inhibit, Shift, Emotional Control, Initiate, Working Memory, Plan/Organize, Organization of Materials, and Monitor. The first three subscales form the Behavioral Regulation Index (BRI), and the last five subscales form the Metacognition Index (MI), and the BRI and MI are then summed to create a Global Executive Composite (GEC) score. The Global Executive Composite (GEC) t-score was used in analyses with higher scores signifying poorer executive function (Mean = 50; Standard Deviation = 10).”

  1. More information should be provided in this section so that readers may get an idea of where the authors have utilized SPSS and where R.

Response: We have provided more context about how SPSS and R were utilized. The section now reads:

“The statistical package SPSS was utilized for data cleaning and management. The statistical package R was utilized to analyze all of the data.”

  1. Table 1 & 2 Which statistical test was used to calculate the reported p-values?

Response: We apologize that this information was missing. We have now added to the Note section of each table to indicate that chi-square tests (Table 1) and Welch’s independent samples t-tests (Table 2) were used.

  1. Figure 1. The percentages within the bars are very small, and the authors must increase their font size.

Response: We have increased the font size for the percentages in the bar graph.

Round 2

Reviewer 2 Report

I think this paper  has been sufficiently improved to warrant publication in Children.